# Investigating the Regulation of Neural Differentiation and Injury in PC12 Cells Using Microstructure Topographic Cues

**DOI:** 10.3390/bios11100399

**Published:** 2021-10-16

**Authors:** Xindi Sun, Wei Li, Xiuqing Gong, Guohui Hu, Junyi Ge, Jinbo Wu, Xinghua Gao

**Affiliations:** 1Materials Genome Institute, Shanghai University, Shanghai 200444, China; sunxindi@shu.edu.cn (X.S.); liwei19981126@163.com (W.L.); gongxiuqing@shu.edu.cn (X.G.); junyi_ge@t.shu.edu.cn (J.G.); 2Shanghai Key Laboratory of Mechanics in Energy Engineering, Shanghai Institute of Applied Mathematics and Mechanics, School of Mechanics and Engineering Science, Shanghai University, Shanghai 200072, China; ghhu@staff.shu.edu.cn

**Keywords:** microfluidics, microstructure topography, PC12 cells, neural differentiation

## Abstract

In this study, we designed and manufactured a series of different microstructure topographical cues for inducing neuronal differentiation of cells in vitro, with different topography, sizes, and structural complexities. We cultured PC12 cells in these microstructure cues and then induced neural differentiation using nerve growth factor (NGF). The pheochromocytoma cell line PC12 is a validated neuronal cell model that is widely used to study neuronal differentiation. Relevant markers of neural differentiation and cytoskeletal F-actin were characterized. Cellular immunofluorescence detection and axon length analysis showed that the differentiation of PC12 cells was significantly different under different isotropic and anisotropic topographic cues. The expression differences of the growth cone marker growth-associated protein 43 (GAP-43) and sympathetic nerve marker tyrosine hydroxylase (TH) genes were also studied in different topographic cues. Our results revealed that the physical environment has an important influence on the differentiation of neuronal cells, and 3D constraints could be used to guide axon extension. In addition, the neurotoxin 6-hydroxydopamine (6-OHDA) was used to detect the differentiation and injury of PC12 cells under different topographic cues. Finally, we discussed the feasibility of combining the topographic cues and the microfluidic chip for neural differentiation research.

## 1. Introduction

The nervous system consists of the peripheral nervous system (PNS), supporting communication between the body and the brain, and the central nervous system (CNS), integrating information from various organ systems [1]. The CNS contains a variety of neuronal cell types connected by axons and dendrites that together build complex neural networks through migration, differentiation, and synapse formation, and realize complex brain functions including transmission, storage, and processing of information, thereby controlling behavior. The main causes of CNS diseases are neurodegeneration and injury [2]. The most common neurodegenerative diseases include Parkinson’s disease (PD), Alzheimer’s disease (AD), and Huntington disease (HD), and their prevalence increases with age [3,4,5,6]. At present, there is no effective treatment for these disorders and their prognosis is poor, seriously impacting the patient’s quality of life and posing a significant socioeconomic burden to society [7,8]. In order to promote neuroregeneration and recover nervous system function, scientists are studying nerve cell growth in vitro. A common strategy is to create structures that support nerve cell growth and axon extension. However, the nervous system has a complex 3D environment consisting of distinct physical, chemical, and biological properties [9,10]. Topographic cues in the physical microenvironment can influence nerve cells, so it is critical to simulate nerve regeneration in vitro [11]. When neurons are injured, the effective connections maintaining the normal function of these cells are lost, and external topographic cues promoting the growth of neurites are necessary to enable the reconstruction of neural connections. Simulation of topographical cues in vitro allows neurons to be placed in a simplified, controlled environment to study the response of neurons to different molecular and physical guidance cues. The different responses of nerve cells to topographical cues include cell adhesion, neurite outgrowth, and cell morphology changes [12,13,14]. These responses can further affect the growth, differentiation, and proliferation of cells in the process of neuroregeneration.

At present, 2D cultures are commonly used although they cannot fully simulate the complex 3D microenvironment of the brain [15,16]. Microfluidics is an emerging fluid control technology that integrates basic operating units involved in the traditional biochemical laboratory on a small chip and thereby enables many analytical functions on a tiny platform. It offers the advantages of convenient operation, savings on reagent costs, and increased analysis speed and has been widely used in many interdisciplinary fields such as biology, chemistry, materials, medicine [17,18,19]. It is worth noting that due to the design of the specific channels of the microfluidic chip, the physical separation of the soma and axon of neurons can be realized for the study of neuronal behavior in a 3D environment [20]. Under the specific microstructure, the cell bodies of nerve cells were fixed in a certain position, and axon behavior could be observed after axons separated the cell bodies. It was convenient to study the behavior of neurons in a specific environment. Francisco et al. studied the ability of embryonic chicken primary sensory neurons to extend axons in a variety of environments characterized by 3D physical constraints in the form of “rooms” and corridor “walls” [21]. Kywe Moe et al. designed a multi-architecture chip (MARC) with different aspect ratios and hierarchical structures, nanometer-to-micrometer sizes, and different structural complexity of the topography. MARCs were then replicated in polydimethylsiloxane (PDMS) to study the effects of different geometric sizes on neural differentiation in primary mouse neural progenitor cells (MNPC) [22]. These studies suggest that microfluidic chips may aid neurological disease models by simulating topographical clues for drug screening or toxicology studies.

Given the above, we designed and manufactured a series of different microstructure topographical cues for inducing neuronal differentiation of cells in vitro, with different topography, sizes, and structural complexities. The different topological microstructures were replicated in Polydimethylsiloxane (PDMS). Replicated microstructures were treated with Poly-L-lysine (PLL). PC12 cells were cultured in their microstructures and then induced neural differentiation using nerve growth factor (NGF). PC12 cells were commonly used for in vitro models of neurodegenerative diseases [23]. Cytoskeletal F-actin and β-tubulin III of neural differentiation were characterized. The results of the cell immunofluorescence experiment and axon length analysis showed that PC12 cells differentiated significantly under different topographic cues. We also explored whether there were differences in the expression of sympathetic nerve marker tyrosine hydroxylase (TH) and growth cone marker growth-associated protein 43 (GAP-43) genes under different topographic cues. Our experimental results showed that the physical environment had an important influence on the differentiation of neuronal cells, and three-dimensional constraints could be used to guide the extension of axons. The neurotoxin 6-hydroxydopamine (6-OHDA) was used to detect the differentiation and injury of PC12 cells under different topographic cues. Finally, we discussed the feasibility of combining the topographic cues and the microfluidic chip for neural differentiation research.

## 2. Materials and Methods

### 2.1. Design and Preparation of Topological Microstructures

Topological microstructures of different patterns were designed using CAD software. In the field area of 8 mm × 2.5 mm, several patterned topographies were designed. Anisotropic topographies include: (1) a straight line of a width of 10 µm, spacing of 10 µm (named S); (2) 45° polyline with a width of 10 µm, spacing of 10 µm (named C45); (3) 90° polyline with a width of 10 µm, spacing of 10 µm (named C90); (4) 135° polyline with a width of 10 µm, spacing of 10 µm (named C135). Isotropic topographies include: (5) 60° distribution micropillar array with a column diameter of 10 µm (named P60); (6) 90° distribution micropillar array with a diameter of 10 µm (named P90). All chambers were 10 µm high. The microstructures were manufactured in silicon using dry deep etching, courtesy of the Shanghai Institute of Microsystems and Information Technology, Chinese Academy of Sciences. 

Different topographic microstructures were replicated using PDMS in subsequent experiments as the substrate for cell adhesion or the bottom layer of the microfluidic chip. The process of replication was as follows: firstly, the silicon template was treated with plasma to remove surface contaminants; then, the silicon template was treated for 1 min using 1H,1H,2H,2H-perfluorooctanyl triethoxy silane (Sigma Aldrich, Burlington, MA, USA) and then heated in a 120 °C circulating blast oven for 2 h. Secondly, PDMS and curing agents were mixed to a ratio of 10:1. The mixture was poured into the silicon template, degassed in a vacuum, and cured in an oven at 65 °C for at least 2 h. Lastly, the PDMS replicas were carefully peeled off the silicon template and sterilized. In order to verify the fidelity of PDMS replicas, they were placed in an oven at 65 °C for 48 h to ensure that the PDMS surface was dry. The obtained PDMS replicas were characterized by SEM (SU8230, Hitachi, Japan).

Microfluidic chip materials include cyclic olefin copolymer (COC), Polymethylmethacrylate (PMMA), PDMS among others [24]. PDMS as a microfluidic chip material shows certain characteristics: high thermal stability, high biocompatibility, good optical properties, chemically inert, and so on [25,26]. Considering the advantages of PDMS, we finally used PDMS as the microfluidic chip material in this experiment. Similarly, there are many types of processing methods for microfluidic systems, such as injection molding, laser ablation, molding, and 3D printing, etc. [27,28]. Each method has its own advantages and disadvantages. For example, 3D printing is flexible and fast in the chip processing process. However, the choice of materials for 3D printing is very limited [29].

### 2.2. Design and Production of the Upper Layer of the Microchip 

The upper layer of the microchip was fabricated using standard soft lithography etching [30]. The upper layer comprised two independent and identical symmetric cell culture channels, each 1000 µm wide and 300 µm high (Figure 1H). The interval between the two cell culture channels was 500 µm, thus forming a cell culture channel limiting nerve cell PC12 to one side to observe the axonal growth. The gap between the two cell culture channels was the main area in which axonal differentiation is observed. SU-8 photoresist was used to fabricate the upper layer of the chip structure. The chip was made of polydimethylsiloxane (PDMS, Sylgard 184, Dow Corning, Midland, MI, USA). The obtained upper layer could be sealed with the patterned bottom layer topological microstructure using oxygen plasma. 

### 2.3. Cell Culture and Seeding

Rat adrenal medullary pheochromocytoma PC12 cells were used in this study. The PC12 cell line was purchased from the Cell Bank/Stem Cell Bank of the Chinese Academy of Sciences Cell (ATCC source). Cells were cultured in Roswell Park Memorial Institute Medium 1640 (1640, Gibco, Waltham, MA, USA) with 10% horse serum (HS, Gibco), 5% fetal bovine serum (FBS, Gibco), and 1% penicillin-streptomycin double antibody (Gibco). Cells were passaged once or twice a week and used between 4 and 6 generations. The obtained PDMS replicas with microstructures were placed in a 6-well plastic culture dish, coated overnight with 0.1% polylysine (PLL, Sigma), and rinsed with phosphate buffer solution (PBS) 3 times. PC12 cells were seeded at a density of 5 × 10^6^ cells/mL and allowed to settle and attach to the surface. After the overnight culture, cells were rinsed with medium to remove any unattached cells. PC12 cells were then cultured in a fresh medium supplemented with 100 ng/mL NGF (BBI) at 37 °C and 5% CO_2_ for 5 days. The medium was changed every 2–3 days. The experimental process of 6-OHDA group was PC12 cells cultured in normal fresh medium for 4 days and treated with 20 ng/mL 6-OHDA for 24 h, and the experimental process of NGF+6-OHDA group was PC12 cells were cultured in fresh medium supplemented with 100 ng/mL NGF for 4 days, then treated with 6-OHDA at a concentration of 20 ng/mL for 24 h.

### 2.4. Immunofluorescence and Imaging

Alexa Fluor^®^ 488 phalloidin (Life Invitrogen, Waltham, MA, USA) was used to characterize cellular F-actin. Staining was performed according to the manufacturer’s instructions. In addition, the tubulin expression of PC12 cells was detected using immunofluorescence staining. Cells on PDMS were washed carefully with phosphate-buffered saline (PBS) and fixed with 4% paraformaldehyde (Sigma) for 20 min. After the addition of goat blocking buffer (Abcam, San Jose, CA, USA), the culture was incubated at room temperature for 1 h. The primary antibody (Tubulin, Abcam), diluted to a ratio of 1:100 (Abcam), was added. The sample was left overnight at 4 °C. After washing with PBS, the corresponding secondary antibody marked with Alexa Fluor 568 was added (goat anti-rabbit IgG). The sample was incubated at room temperature for 1 h. After the addition of 4′,6-diamidino-2-phenylindole (DAPI, Life Invitrogen) to counterstain nuclei, the sample was incubated at room temperature for 10 min. After washing with PBS three times, fluorescence images were acquired.

### 2.5. Total RNA Isolate and RT-qPCR 

We separated total RNA from the F group, S group, and P 90 group of PC12 after treated NGF 5 days with TRIzol^®^ reagent according to the manufacturer’s instructions (Invitrogen). We used a One-Step RT-qPCR Kit (Sangon Biotech, Shanghai, China) to perform cDNA synthesis and PCR reaction on total RNA and used Lightcycle 96 (Roche, Basel, Switzerland) for detection. Average CT values were normalized to GAPDH expression levels. Primers synthesized by Sangon Biotech (Shanghai) Co., Ltd. were as follows: 

GAP-43: 5′-CGACAGGATGAGGGTAAAGAA-3′ (forward), 5′-GACAGGAGAGGAAACTTCAGAG-3’ (reverse); 

TH: 5′-GTGAACCAATTCCCCATGTG-3′ (forward), 5′-CAGTACCGTTCCAGAAGCTG-3′ (reverse); 

GAPDH: 5′-TCCAGTATGACTCTACCCACG-3′ (forward), 5′-CACGACATACTCAGCACCAG-3′ (reverse).

### 2.6. Data Analysis

Image J was used for image analysis and cell counting. CellSens Standard software was used for the axon length measurement. The standard errors in the experimental data were obtained from Student’s *t*-test under multiple groups of data (*n* ≥ 3).

## 3. Results and Discussion

### 3.1. Characterization of Topological Microstructures in PDMS Replicas

The PDMS replicas with topological microstructures were obtained from the silicone templates and patterns included isotropic pillars of different sizes and anisotropic micrometer gratings of different angles. The patterning groups could be divided into anisotropic and isotropic topographies, of which anisotropic topographies were the straight channel (S), 45° polyline (C45), 90° polyline (C90), and 135° polyline (C135), and isotropic topographies were the 60° distribution micropillar array (P60), and 90° distribution micropillar array (P90). All patterns were characterized by SEM as shown in Figure 1A–G. There are a total of 6 patterned topographic microstructures and flat surfaces (F). Topological microstructures of the PDMS replicas exhibited good fidelity with a smooth surface and clear structure. Figure 1H showed the optical photo of the upper layer mold.

### 3.2. Differentiation of PC12 Cells in Different Topological Microstructures

The pheochromocytoma cell line (PC12) is a validated neuronal cell model that is widely used to study neuronal differentiation [22]. PC12 cells rely on nerve growth factor (NGF) and respond to this neurotrophic factor with nerve cell differentiation, showing a typical neural phenotype and neurite outgrowth. In this experiment, PC12 cells were cultured in different topological microstructures together with nerve growth factor NGF at a concentration of 100 ng/mL to establish a microchip neuronal differentiation model. In response to NGF, PC12 cells proliferated and showed obvious neurite growth, suggesting that the in vitro differentiation model was successfully established. Cellular immunofluorescence images visualizing F-actin and neuronal-associated protein β-tubulin III are shown in Figure 2 and Figure 3. 

We analyzed the axonal length of the PC12 cells on different micro topologies, similar to previous work by Spillane et al. [31]. The histogram of the axon length of PC12 cells on different micro topologies at day 5 was shown in Figure 4A,B. The results show that when PC12 cells were not treated with NGF, the average axon length under different conditions was all less than 10 µm, which was much smaller than the result of PC12 cells treated with NGF. And then the morphology of PC12 cells on different micro topologies changed significantly after applying NGF treatment. This indicated that NGF played a key role in the axon elongation and neural differentiation of PC12 cells. And the results also show that PC12 cells grown on different microstructures could differentiate into neurons and exhibited increased cell body size, sprouting neurites, and the formation of a neural network. In this, compared with the flat group (F group), the axon length of PC12 cells increased significantly in the S, P60, P90, and C135 groups. The C45 and C90 groups may be affected by the bending angle of the polyline microstructure, the axon length did not become significantly longer, and C45 was also significantly shorter and only 20.3 ± 2.8 µm. This revealed that the axon could rotate around the corner within the range of 45° to 180° of the measured angles on anisotropic patterns; the larger this angle, the more pronounced the axon growth was (S: 84.0 ± 2.9 µm > C135: 54.8 ± 3.8 µm > C90: 48.1 ± 2.5 µm > C45: 20.3 ± 2.8 µm).

In addition, PC12 cell differentiation also showed a certain cell orientation and the ability to form neural networks. The axon extension on C45, C90, C135, and S structures tended to occur in the direction of the topography itself. Conversely, the neurite extension on P60 and P90 structures was relatively more random, although most of the axons turned and extended around a pillar. The immunofluorescence staining results of β-tubulin (Figure 3) and F-actin were similar. Among them, the cell axons of the S group showed the orientation of growing along the straight microstructure, while in the P60 and P90 group the cells were similar to the F group and could form the neural network. It might be because there was much space between the micropillars in the P60 and P90 groups the physical constraints have little effect on the direction of cell axon growth. This was also reflected in the similar axon length of P60 (68.3 ± 4.1 µm) and P90 (63.8 ± 3.6 µm) and suggested that the axon length was not affected by the angle between the micropillars.

To further prove the relationship between microstructural cues and PC12 neural differentiation, the expressions of growth cone marker GAP-43 and sympathetic marker TH were used to identify the sympathetic characteristics of NGF-treated PC12 cells in different topographies, similar to previous work by Li-Ying Zhong et al. [32]. According to the previous experimental results, we chose the anisotropic topography S and the isotropic topography P90. The results were shown in Figure 4C,D. Compared with the control group (F group), the mRNA expression of the GAP-43 gene and TH gene in S and P90 topographies were significantly increased (*p* < 0.01), and the increase in S topography was more significant. This indicated that some special microstructure cues could guide neurons to grow in microenvironments of different terrain, size, and structural complexity, and specific three-dimensional constraints can effectively guide the extension of axons.

### 3.3. Cellular Neurotoxin Injury of PC12 Cells on Different Topological Microstructures

6-OHDA is a neurotoxin with a similar structure to dopamine which can selectively elicit cell death of dopaminergic neurons [33]. It is widely used in animal models of Parkinson’s disease. Figure 5 shows immunofluorescence images of β-tubulin III of PC12 cells treated with 6-OHDA (20 ng/mL) under different conditions. PC12 cells treated with 6-OHDA on different patterned surfaces exhibited substantial cellular apoptosis and almost no axonal differentiation. When the PC12 cells were pretreated with NGF for 5 days to obtain differentiated cells, and then 6-OHDA was added, the cells still exhibited a certain degree of apoptosis, but some axons could have remained. We also calculated the axon length as shown in Figure 6. The effect of 6-OHDA on cells was huge. The axons were hardly generated, and the generated axons also disappear as the cell dies. Among all groups, after being treated by NGF and 6-OHDA, the remaining axon length of the S group was the longest. This result may be helpful for us to construct a nerve injury model or a toxicological drug evaluation model.

### 3.4. Analysis on the Trend of Neuronal Axon Differentiation on a Microchip

Based on the above research, we also discussed the feasibility of combining the topographic cues and the microfluidic chip for neural differentiation research. The design of the combined microchip is shown in Figure 7A. The microchip was contained two parts; one was the upper layer containing two independent channels and the other was the patterned bottom layer with topological microstructure like the S group. PC12 cells were seeded on the left side microchannel and the drug or growth factor was added to the right microchannel. After mixing the drug with the medium and after mixing the growth factor with the medium, pipetting guns were used to slowly add the right side of the microchannel, and the above steps were repeated every 8 h. The microchannels on the left and right sides were connected to each other by the straight microstructure. The results of immunofluorescence and axon length were shown in Figure 7B,C. The results indicated that there was significant differentiation of PC12 cells treated with NGF, with axon lengths reaching 105.8 ± 6.1 μm. This result was even greater than the axon length of PC12 on the S microstructure cues with NGF (84.0 ± 2.9 µm). In addition, the axon orientation of the cells was obvious, and they all extend along the microstructure topography to the microchannel on the other side. On the other hand, the neurotoxin experiment with 6-OHDA added also showed similar results. PC12 cells died to a certain extent after 6-OHDA treatment. PC12 cells undergo 6-OHDA treatment after differentiation, some axons of PC12 cells remained, but there was serious damage. We believe that the combined model of the microchip with microstructure cues may be more conducive to the observation of cell neural differentiation and the measurement of axon size. It can simulate related neurological disease models in vitro. 

## 4. Conclusions

To summarize, here we describe the design and manufacture of a series of different topographical cues for the culture of PC12 cells in vitro. The structures enabled guidance of neuronal growth in distinct 3D microenvironments with different topographies, sizes, and structural complexity. We studied the neuronal differentiation of PC12 cells in these topologies using PDMS replicas and assessed the influence of biophysical factors on the differentiation of nerve cells. Using these topologies, we can effectively simulate the 3D environment of the human body in vitro and show that 3D constraints could be used to guide axonal extension. This research using a microfluidic chip platform is a small part of the potential application of neuronal network reconstruction in vitro. Future work may increase the complexity of the microchannel, involving more complex networks of more neuronal subtypes and provide the basis for the in vitro study of Alzheimer’s disease, Parkinson’s disease, and other neurological disorders.

## Figures and Tables

**Figure 1 biosensors-11-00399-f001:**
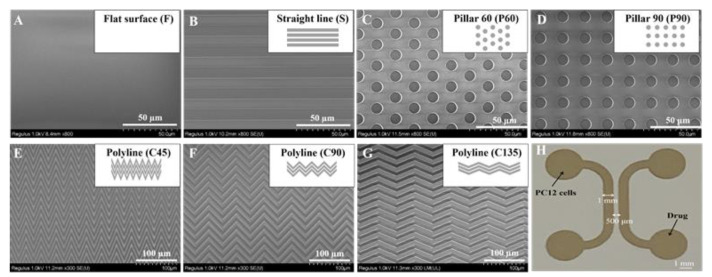
SEM images of PDMS replicas with different topological microstructures and images of the upper layer with the symmetric cell culture channels. (**A**) flat surface (F). (**B**) straight channel (S). (**C**) 60° distribution micropillar array (P60). (**D**) 90° distribution micropillar array (P90). (**E**) 45° polyline (C45). (**F**) 90° polyline (C90). (**G**) 135° polyline (C135). Scale bar = 50 µm or 100 µm. (**H**) Images of the upper layer mold with the symmetric cell culture channels. Scale bar = 1 mm.

**Figure 2 biosensors-11-00399-f002:**
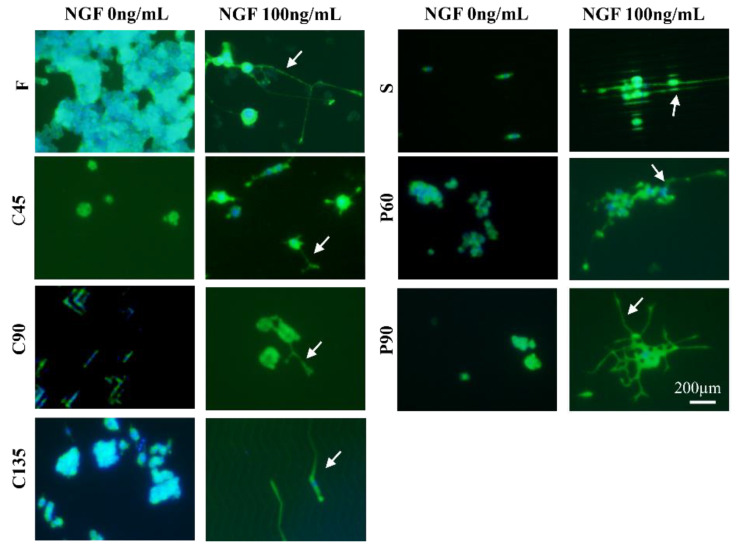
Fluorescence images of F-actin expression of PC12 cells under different conditions. Green, F-actin; blue, cell nucleus; NGF:100 ng/mL; Scale bar = 200 µm.

**Figure 3 biosensors-11-00399-f003:**
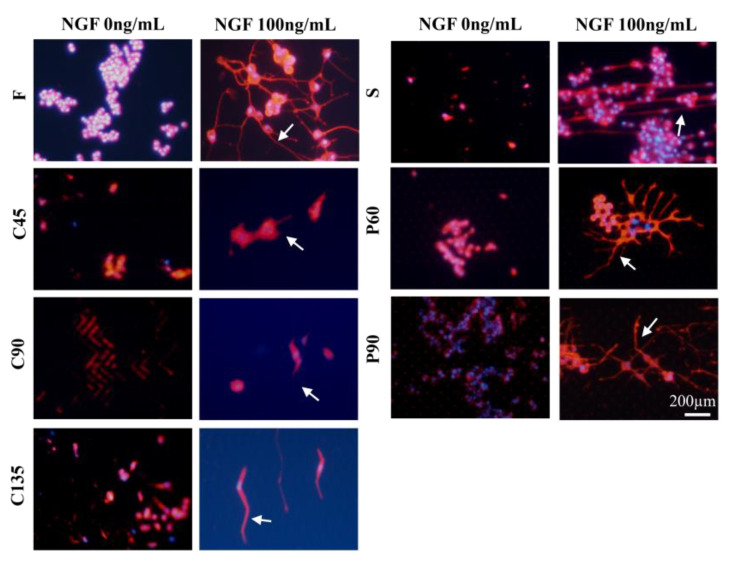
Fluorescence images of β-tubulin expression of PC12 cells under different conditions. Red, β-tubulin; blue, cell nucleus; NGF:100 ng/mL; Scale bar = 200 µm.

**Figure 4 biosensors-11-00399-f004:**
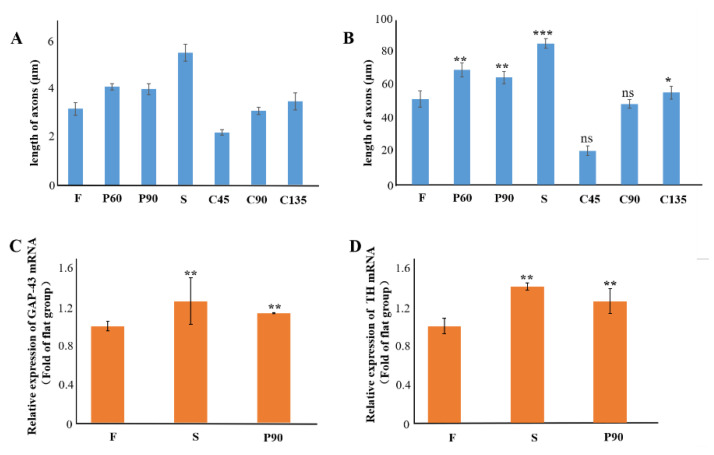
(**A**) Histogram of axon length of PC12 cell on different topographies after 5 days. (**B**) Histogram of axon length of PC12 cell on different topographies after NGF treatment for 5 days. (**C**) Gene expression of GAP-43 of PC12 cell on different topographies after NGF treatment for 3 days. (**D**) Gene expression of TH of PC12 cell on different topographies after NGF treatment for 3 days. NGF: 100 ng/Ml, GAPDH refers to the internal parameter, * *p* < 0.05, ** *p* < 0.01, *** *p* < 0.001. The data is the mean value of each parameter for each experimental condition.

**Figure 5 biosensors-11-00399-f005:**
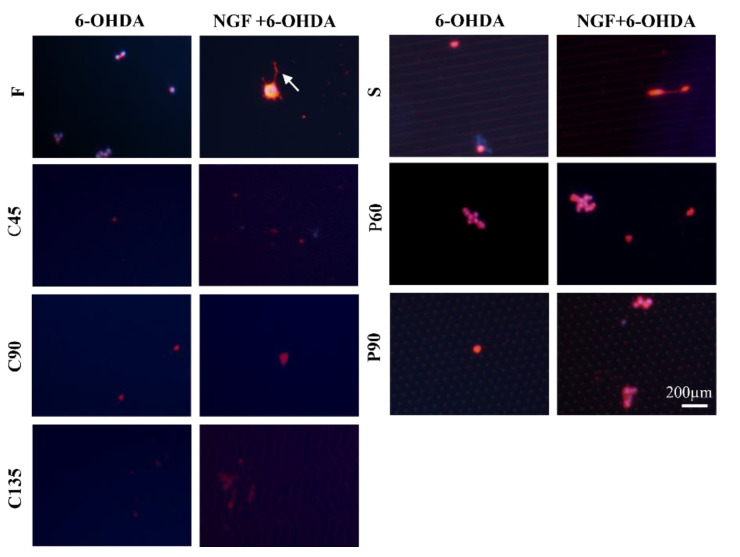
Fluorescence images of β-tubulin expression of PC12 cells under different conditions after 6-OHDA treatment. Red, β-tubulin; blue, cell nucleus; NGF: 100 ng/mL; 6-OHDA: 20 ng/mL. Scale bar = 200 µm.

**Figure 6 biosensors-11-00399-f006:**
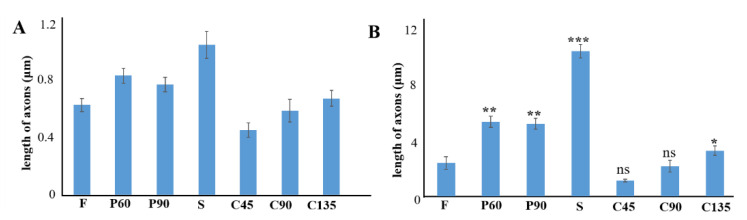
**(****A**) Histogram of axon length of PC12 cell on different topographies after 6-OHDA treatment without NGF. (**B**) Histogram of axon length of PC12 cell on different topographies after 6-OHDA and NGF treatment. NGF: 100 ng/mL. 6-OHDA: 20 ng/mL. * *p* < 0.05, ** *p* < 0.01, *** *p* < 0.001. The data was the mean value of each parameter for each experimental condition.

**Figure 7 biosensors-11-00399-f007:**
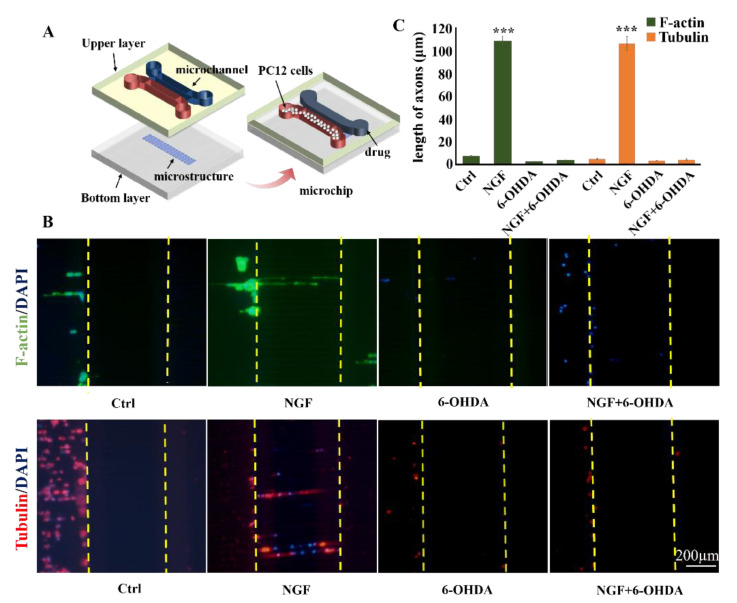
(**A**) Chip design. (**B**) Fluorescence images of F-actin and β-tubulin expression of PC12 cells on the combined chip after 5 days. Red, β-tubulin; blue, cell nucleus; Scale bar = 200 µm. C: Histogram of axon length of PC12 cell on the combined chip after 5 days. NGF: 100 ng/mL; 6-OHDA: 20 ng/mL. *** *p* < 0.001. The data is the mean value of each parameter for each experimental condition.

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
