# Peer review of "Investigating the Regulation of Neural Differentiation and Injury in PC12 Cells Using Microstructure Topographic Cues"

_biosensors, 2021, doi:10.3390/bios11100399_

Round 1

Reviewer 1 Report

Gao et al. report in this paper on the study on a family of different topographical microstructures for inducing neuronal differentiation of pheochromocytoma cells (PC12) in vitro. The microstructures allowed, at least in part, the 3D neuronal growth with some previously designed constraints to guide the axon extension. The authors claim that the physical environment has an important effect on the differentiation of PC12 cells and that the 6-hydroxydopamine molecule detects the differentiation and injury of such cells. Overall, the research work is OK and reaches the standard of the journal Biosensors. So that, my recommendation is to publish the work, but after the following minor revisions.

  1. In page 1, the “Citation” section should be completed.

  2. The authors should define what PC12 are in both Abstract and Introduction sections.

  3. In page 3, line 113, please changed “...PDMS replicas was...” by “...PDMS replicas were...”.

  4. The main text must be 'justified' from page 3, that is, from “2.3 Cell culture and seeding“ point on.

Author Response

Dear reviewer,

Thank you very much for your comments on the manuscript. The manuscript has been revised and some modifications have been made according to the comments. The responses to each point are listed as following.

Sincerely,

  Jinbo Wu and Xinghua Gao  

Response to Reviewer 1

Point 1: In page 1, the “Citation” section should be completed.

Response 1: Thanks for your comments. The manuscript has been modified.

In page 1, line 24-28, we have added “Sun X.; Li W.; Gong X.; Hu G.; Ge J.; Wu J.; Gao X.. Investigating the regulation of neural differentiation and injury in PC12 cells using microstructure topographic cues. Biosensors 2021, 11, x. https://doi.org/10.3390/xxxxx” in the manuscript.

Point 2: The authors should define what PC12 are in both Abstract and Introduction sections.

Response 2: Thanks for your comments. PC12 have been defined in both Abstract and Introduction sections.

In page 1, line 12-13, we have added “The pheochromocytoma cell line PC12 is a validated neuronal cell model that is widely used to study neuronal differentiation.” in the manuscript.

In page 2, line 77-78, we have added “PC12 cells were commonly used for in vitro models of neurodegenerative diseases [22].” in the manuscript.

Point 3:In page 3, line 113, please changed “...PDMS replicas was...” by “...PDMS replicas were...”.

Response 3: Thanks for your comments. The manuscript has been modified.

In page 3, line 112, we have changed “...PDMS replicas was...” to “...PDMS replicas were...”.

Point 4:The main text must be 'justified' from page 3, that is, from “2.3 Cell culture and seeding” point on.

Response 4: Thanks for your comments. The manuscript has been modified.

From “2.3 Cell culture and seeding” point on, the main text have been justified.

Reviewer 2 Report

With pleasure, I read the manuscript “Investigating the regulation of neural differentiation and injury in PC12 cells using microstructure topographic cues”. The document presents the design, microfabrication and characterization of different microstructures for inducing neuronal differentiation of PC12 cells in vitro, using nerve growth factor, and explored the effect of the physical environment in that process. This paper is interesting and has relevance for the readers, with clear methods and conclusions supported by the achieved results. It is well structured and clear. I have, however, some comments and suggestions to improve the manuscript.

  • Page 2, Line 59-62: “It is worth noting that due to the design of the specific channels of microfluidic chip, the physical separation of the soma and axon of neurons can be realized for the study of neuronal behavior in a 3D environment”. This sentence is confusing. Are the authors referring to specific geometries for cells separation in planar microfluidics? What is the advantage of achieving separation for this application? Please clarify.
  • Page 3, line 106 – The PDMS structures were fabricated through replica molding using silicon molds. Were these molds produced by dry deep etching, similarly to the first silicon microstructures? The manuscript would benefit if the authors added a scheme of figure detailing the step-by-step fabrication process, both for the microstructures and for the upper layer.
  • Page 3, line 121: please add a photo of the upper layer with the symmetric cell culture channels.
  • Page 3, line 128 – how did the authors align both PDMS layers to assure the correct positioning of cell culture channels and topographic microstructures?
  • Page 4, section 2.6: data analysis – the authors should include additional details regarding data analysis in imageJ, such as the functions and parameters (when relevant) that were used or cell counting and image analysis.
  • In Figure 1 (line 197), it would be valuable to add also the SEM image of the flat surface, to show the expected smooth surface.
  • Page 7, line 258 – why did the authors choose, for this study, the topographies S and P90 and not P60 (which had better results than P90)?
  • Figure 7, in particular the plot C should be bigger to be easier to observe the presented data.
  • In section 3.4, where the authors present the integrated microchip, the drugs and growth factor were added to one of the ramifications of the microchannel (and the cells were seeded on the other side). What flow rate or fluid velocity was used here to pump the drugs and growth factor? This discussion is important since, in microfluidics, the flow rate has an enormous effect on the diffusion and mass transport and, consequently, a different flow rate could still improve the neural differentiation. Did the authors evaluate this effect? Please comment.
  • The document should be reviewed by an English native speaker. There are some gramar errors and some sentences are confusing. Examples of errors to be corrected: page 2, line 76 – “microstructures were replicated by Polydimethylsiloxane (PDMS).” Should be “replicated in Polydimethylsiloxane (PDMS).” ; Page 3, line 113 – “PDMS replicas was” should be “replicas were”.

Author Response

Dear reviewer,

Thank you very much for your comments on the manuscript. The manuscript has been revised and some modifications have been made according to the comments. The responses to each point are listed as following.

Sincerely,

  Jinbo Wu and Xinghua Gao  

Response to Reviewer 2 

Point 1:Page 2, Line 59-62: “It is worth noting that due to the design of the specific channels of microfluidic chip, the physical separation of the soma and axon of neurons can be realized for the study of neuronal behavior in a 3D environment”. This sentence is confusing. Are the authors referring to specific geometries for cells separation in planar microfluidics? What is the advantage of achieving separation for this application? Please clarify.

Response 1: Thanks for your comments. Under the specific microstructure, the cell bodies of nerve cells were fixed in a certain position, and axon behavior could be observed after axons separated the cell bodies. It was convenient to study the behavior of neurons in a specific environment. This sentence comed from "Notably, microflfluidic devices allowing the physical separation of soma from axons, thanks to micro-channels, were used to study neuronal cell behavior" in Ref.20.

Point 2:Page 3, line 106 – The PDMS structures were fabricated through replica molding using silicon molds. Were these molds produced by dry deep etching, similarly to the first silicon microstructures? The manuscript would benefit if the authors added a scheme of figure detailing the step-by-step fabrication process, both for the microstructures and for the upper layer.

Response 2: Thanks for your comments. These molds of microstructures were provided by Shanghai Institute of Microsystems and Information Technology, Chinese Academy of Sciences. They can not provide the preparation method. The mold of upper layer was fabricated using standard soft lithography etching. Refer to Ref.30 for details. 

Point 3:Page 3, line 121: please add a photo of the upper layer with the symmetric cell culture channels.

Response 3: Thanks for your comments. The manuscript has been modified.

In page 5, line 207, we have added a photo of the upper layer with the symmetric cell culture channels in figure 1-H.

Point 4:Page 3, line 128 – how did the authors align both PDMS layers to assure the correct positioning of cell culture channels and topographic microstructures?

Response 4: Thanks for your comments. The size of the topographic microstructures replicated in PDMS was larger than the 500 μm width of the cell culture channel, which facilitated alignment and manipulation when cell culture channels and topographic microstructures were sealed.

Point 5:Page 4, section 2.6: data analysis – the authors should include additional details regarding data analysis in imageJ, such as the functions and parameters (when relevant) that were used or cell counting and image analysis.

Response 5: Thanks for your comments. The manuscript has been modified. 

In page 4, section 2.6,we have added“Image J was used for image analysis and cell counting. CellSens Standard software was used for the axon length measurement. The standard errors in the experimental data were obtained from Student's t-test under multiple groups of data (n ≥ 3).”.

Point 6:In Figure 1 (line 197), it would be valuable to add also the SEM image of the flat surface, to show the expected smooth surface.

Response 6: Thanks for your comments. The manuscript has been modified. 

In page 5,line 207, we have added SEM image of the flat surface.

Point 7:Page 7, line 258 – why did the authors choose, for this study, the topographies S and P90 and not P60 (which had better results than P90)?

Response 7: Thanks for your comments. Compared with other anisotropic topographies, the axon length of S was the longest. So we choose topography S on anisotropic topographies for this study. The axon length of P60 (68.3±4.1 µm) and P90 (63.8±3.6 µm) on isotropic topographies were similar; This result suggests that the axon length was not affected by the angle between the micropillars. So we randomly choose topography P90 on isotropic topographies for mRNA expression of growth cone marker GAP-43 and sympathetic marker TH.

Point 8:Figure 7, in particular the plot C should be bigger to be easier to observe the presented data.

Response 8: Thanks for your comments. The manuscript has been modified. We have updated the new versions of Figure 7.

Point 9:In section 3.4, where the authors present the integrated microchip, the drugs and growth factor were added to one of the ramifications of the microchannel (and the cells were seeded on the other side). What flow rate or fluid velocity was used here to pump the drugs and growth factor? This discussion is important since, in microfluidics, the flow rate has an enormous effect on the diffusion and mass transport and, consequently, a different flow rate could still improve the neural differentiation. Did the authors evaluate this effect? Please comment.

Response 9: Thanks for your comments. In this experiment, after mixing drug with the medium and after mixing the growth factor with the medium, pipetting guns were used to slowly add one side of the microchannel (the cells were seeded on the other side), and the above steps were repeated every 8 hours. So there was no discussion about the impact of flow on diffusion and mass transport.

Point 10:The document should be reviewed by an English native speaker. There are some gramar errors and some sentences are confusing. Examples of errors to be corrected: page 2, line 76 – “microstructures were replicated by Polydimethylsiloxane (PDMS).” Should be “replicated in Polydimethylsiloxane (PDMS).” ; Page 3, line 113 – “PDMS replicas was” should be “replicas were”.

Response 10: Thanks for your comments. The manuscript has been modified.

In page 2, line 74 “The different topological microstructures were replicated in Polydimethylsiloxane(PDMS) ”;In page 3, line 112 “Lastly, the PDMS replicas were carefully peeled off the silicon template and sterilized”. Other errors have been also modified in manuscript.

Reviewer 3 Report

The following comments may help the authors to improve the manuscript before an acceptance.

  1. The authors did not explicitly mention why they choose PDMS for the chip material.
  2. PDMS is not an ideal material for commercialization or scaling up the experiments as it is expensive. There are some other materials for chip-making such as COC, COP, PMMA, etc. See reference: 10.1039/C9LC00888H (Paper) Lab Chip, 2019, 19, 3825-3833

The fabrication method of the master mould normally requires cleanroom and expensive photolithography. Please discuss some alternative fabrication methods (see the aforementioned reference) such as injection moulding, micromilling.

3. Font sizes in the figures such as figure 7 are so small. Please enlarge them.

Author Response

Dear reviewer,

Thank you very much for your comments on the manuscript. The manuscript has been revised and some modifications have been made according to the comments. The responses to each point are listed as following.

Sincerely,

  Jinbo Wu and Xinghua Gao  

Response to Reviewer 3 

Point 1:The authors did not explicitly mention why they choose PDMS for the chip material.

Response 1: Thanks for your comments. The manuscript has been modified.

In page 3, line 117-122,we have added “PDMS as a microfluidic chip material shows certain characteristics: good insulation; high thermal stability; high biocompatibility; good optical properties; non-toxic, durable, chemically inert, short production cycle; flexible packaging methods and so on[24-25]. Considering the advantages of PDMS, we finally used PDMS as the microfluidic chip material in this experiment.” in the manuscript.

Point 2PDMS is not an ideal material for commercialization or scaling up the experiments as it is expensive. There are some other materials for chip-making such as COC, COP, PMMA, etc. See reference: 10.1039/C9LC00888H (Paper) Lab Chip, 2019, 19, 3825-3833.The fabrication method of the master mould normally requires cleanroom and expensive photolithography. Please discuss some alternative fabrication methods (see the aforementioned reference) such as injection moulding, micromilling.

Response 2: Thanks for your comments. Microfluidic chip materials include COC, PMMA, PDMS and so on. PDMS as a microfluidic chip material shows certain characteristics: good insulation; high thermal stability; high biocompatibility; good optical properties; non-toxic, durable, chemically inert, short production cycle; flexible packaging methods and so on .In addition, according to references "Optimising the supercritical angle fluorescence structures in polymer microfluidic biochips for highly sensitive pathogen detection: a case study on Escherichia coli"、"Latest developments in micro total analysis systems"、"Fabrication of a flexible multi-referenced surface plasmon sensor using room temperature nanoimprint lithography", the biocompatibility, chemically inert and stability of PDMS were taken into account. Therefore, PDMS was finally used as the microfluidic chip material in this experiment.

Similarly, there are many types of processing methods for microfluidic systems, such as injection molding, laser ablation, molding, micro-nano imprinting, and 3D printing, etc. Each method has its own advantages and disadvantages. For example, 3D printing is flexible and fast in the chip processing process, and it can also complete the sealing of the fluid channel during the printing process, which improves the processing efficiency. However, the choice of materials for 3D printing is very limited. In addition, according to references "High-precision modular microfluidics by micromilling of interlocking injection-molded blocks"、"μOrgano: A Lego-Like Plug & Play System for Modular Multi-Organ-Chips"、"A Modular Microfluidic Device via Multimaterial 3D Printing for Emulsion Generation", In this experiment, considering the fidelity and resolution of various graphics, combined with the existing experimental conditions of the research group, we chose dry etching microstructures mask and wet etching upper layer mask.

In page 3, line 117-128, we have added “Microfluidic chip materials include COC, PMMA, PDMS and so on [23]. PDMS as a microfluidic chip material shows certain characteristics: good insulation; high thermal stability; high biocompatibility; good optical properties; non-toxic, durable, chemically inert, short production cycle; flexible packaging methods and so on [24-25]. Considering the advantages of PDMS, we finally used PDMS as the microfluidic chip material in this experiment. Similarly, there are many types of processing methods for microfluidic systems, such as injection molding, laser ablation, molding, micro-nano imprinting, and 3D printing, etc. [26-27]. Each method has its own advantages and disadvantages. For example, 3D printing is flexible and fast in the chip processing process, and it can also complete the sealing of the fluid channel during the printing process, which improves the processing efficiency. However, the choice of materials for 3D printing is very limited [28].” in the manuscript.

Point 3Font sizes in the figures such as figure 7 are so small. Please enlarge them.

Response 3: Thanks for your comments. The manuscript has been modified. We have updated the new versions of Figure 7.

Reviewer 4 Report

Different cellular responses to various topographical cues have been more demonstrated in a plethora of literatures, no exception in the field of neural regeneration. This manuscript describes the neural orientational responses of PC12 cells cultured on six different topographical surfaces, involving 4 anisotropic and 2 isotropic, and presenting some phenomenological observations including the axonal length, cell orientation and the ability to form neural networks, and the expressions of growth cone marker GAP-43 and sympathetic marker TH of the PC12 cells on these different microtopologies. However, these results look not very fresh if compared with similar works in this field, such as that of revealed in some review articles ‘Herbert Francisco, Benjamin B. Yellen, Derek S. Halverson, Gary Friedman, Gianluca Gallo: Regulation of axon guidance and extension by three-dimensional constraints, Biomaterials 28 (2007) 3398–3407’, ‘Diane Hoffman-Kim, Jennifer A. Mitchel, and Ravi V. Bellamkonda: Topography, Cell Response, and Nerve Regeneration, Annu. Rev. Biomed. Eng. 2010. 12:203–231’, ‘HoJun Jeon, Carl G. Simon, Jr, GeunHyung Kim: A mini-review: Cell response to microscale, nanoscale, and hierarchical patterning of surface structure, J Biomed Mater Res Part B 2014:102B:1580–1594’, ‘C. Simitzi, A. Ranella, E. Stratakis: Controlling the morphology and outgrowth of nerve and neuroglial cells: The effect of surface topography, Acta Biomaterialia 51 (2017) 21–52’, name a few. Before being able to publish, some questions need further answers: what is your rational to design such six different microtopologies? what is the problem you want to resolve? what is new understanding from the six cases when they are used as sample ‘contrast’ series in your experiments, in comparation with current published papers? What microfluidic parameters influence the related results? and so on. In addition, the English writing of this manuscript needs substantial improvement. That’s obviously not good writing, for example, a large section of the Abstract repeats again in the Introduction; what’s ‘broken line’ in Fig.1? etc..

Author Response

Dear reviewer,

Thank you very much for your comments on the manuscript. The manuscript has been revised and some modifications have been made according to the comments. The responses to each point are listed as following.

Sincerely,

  Jinbo Wu and Xinghua Gao  

Response to Reviewer 4 

Point 1What is your rational to design such six different microtopologies?

Response 1: Thanks for your comments.The purpose of our experiment was to study the effects of different topographic cues on nerve differentiation and injury of PC12 cells, so we designed these six topographic microstructures , including both isotropic topographies(P60、P90) and anisotropic topographies(S、 C45、 C90、 C135).

Point 2what is the problem you want to resolve?

Response 2: Thanks for your comments. The question I want to explore the impact of biophysical factors (biophysical factors in this study are clues of different topographies) on the differentiation of nerve cells. Combine topographic cues with microfluidic chips to study whether neural differentiation was feasible.

Point 3what is new understanding from the six cases when they are used as sample ‘contrast’ series in your experiments, in comparation with current published papers?

Response 3: Thanks for your comments. Compared with current published papers, PC12 cells differentiated significantly under six different topographic cues. We also explored whether there were differences in the expression of sympathetic nerve marker tyrosine hydroxylase (TH) and growth cone marker growth-associated protein 43 (GAP-43) genes under different topographic cues. We have also probed the application in drug and toxicological evaluation.

Point 4What microfluidic parameters influence the related results?

Response 4: Thanks for your comments. Microfluidic parameters influence the related results,such as morphology and size of the microstructure.  

Point 5A large section of the Abstract repeats again in the Introduction.

Response 5: Thanks for your comments. The repetitive parts of the introduction and abstract has been revised in the manuscript.

Point 6what’s ‘broken line’ in Fig.1?

Response 6: Thanks for your comments. We have changed “broken line” to “polyline ” in Fig. 1.

Round 2

Reviewer 2 Report

The authors improved the manuscript when compared to the first submission. However, some comments and concerns were not totally explained. Here follow some topics that should be addressed before the paper can be accepted for publication.

Page 2 – “It is worth noting that due to the design of the specific channels of microfluidic chip, the physical separation of the soma and axon of neurons can be realized for the study of neuronal behavior in a 3D environment [20]”. In the paper, it is not clear yet why the authors want this separation. For instance, in the response letter, the authors state that: “Under the specific microstructure, the cell bodies of nerve cells were fixed in a certain position, and axon behavior could be observed after axons separated the cell bodies. It was convenient to study the behavior of neurons in a specific environment.” This explanation should be added to the manuscript, in this section.

Page 3 – the authors added a new paragraph where it is stated: “Microfluidic chip materials include COC, PMMA, PDMS and so on”. “And so on” could be replaced by “among others”. Please include the definitions of the acronyms of COC and PMMA.

In figure 1, instead of “bar”, replace by “scale bar”.

In my previous review, I noticed that there was no discussion on the effect of microfluidic parameters, namely flow rates or fluid velocities. According to the response letter, the authors performed the assays manually, using pipettes. Consequently, the fluids flow in the microdevice due to the manually applied pressure, and there is no control of these variables. Indeed, the authors state in the response letter that this was not discussed (and it should be, since modifying microfluidic parameters will significantly alter the performance of the microdevice). I suggest the authors to add in the manuscript the methods explanation presented in the response letter:

“In this experiment, after mixing drug with the medium and after mixing the growth factor with the medium, pipetting guns were used to slowly add one side of the microchannel (the cells were seeded on the other side), and the above steps were repeated every 8 hours.”

Author Response

Dear reviewer,

Thank you very much for your comments on the manuscript. The manuscript has been revised and some modifications have been made according to the comments. The responses to each point are listed as following.

Sincerely,

  Jinbo Wu and Xinghua Gao  

Response to Reviewer 2 

Point 1:Page 2 – “It is worth noting that due to the design of the specific channels of microfluidic chip, the physical separation of the soma and axon of neurons can be realized for the study of neuronal behavior in a 3D environment [20]”. In the paper, it is not clear yet why the authors want this separation. For instance, in the response letter, the authors state that: “Under the specific microstructure, the cell bodies of nerve cells were fixed in a certain position, and axon behavior could be observed after axons separated the cell bodies. It was convenient to study the behavior of neurons in a specific environment.” This explanation should be added to the manuscript, in this section.

Response 1: Thanks for your comments. The manuscript has been modified.

In page 2, line 61-63,we have added “Under the specific microstructure, the cell bodies of nerve cells were fixed in a certain position, and axon behavior could be observed after axons separated the cell bodies. It was convenient to study the behavior of neurons in a specific environment.”in the manuscript.

Point 2:Page 3 – the authors added a new paragraph where it is stated: “Microfluidic chip materials include COC, PMMA, PDMS and so on”. “And so on” could be replaced by “among others”. Please include the definitions of the acronyms of COC and PMMA.

Response 2: Thanks for your comments. The manuscript has been modified.

In page 3, line 120-121,we have changed“Microfluidic chip materials include COC, PMMA, PDMS and so on” to “Microfluidic chip materials include cyclic olefin copolymer (COC), Polymethylmethacrylate(PMMA), PDMS among others”

Point 3:In figure 1, instead of “bar”, replace by “scale bar”.

Response 3: Thanks for your comments. The manuscript has been modified.

In page 5, line 211-212,we have changed“bar”to “scale bar”in figure 1.

Point 4:In my previous review, I noticed that there was no discussion on the effect of microfluidic parameters, namely flow rates or fluid velocities. According to the response letter, the authors performed the assays manually, using pipettes. Consequently, the fluids flow in the microdevice due to the manually applied pressure, and there is no control of these variables. Indeed, the authors state in the response letter that this was not discussed (and it should be, since modifying microfluidic parameters will significantly alter the performance of the microdevice). I suggest the authors to add in the manuscript the methods explanation presented in the response letter: “In this experiment, after mixing drug with the medium and after mixing the growth factor with the medium, pipetting guns were used to slowly add one side of the microchannel (the cells were seeded on the other side), and the above steps were repeated every 8 hours.”

Response 4: Thanks for your comments. The manuscript has been modified.

In page 9, line 318-320,we have added“After mixing drug with the medium and after mixing the growth factor with the medium, pipetting guns were used to slowly add the right side of the microchannel, and the above steps were repeated every 8 hours.”.

Reviewer 3 Report

I do not have other comments.

Author Response

Thank you very much for your comments on the manuscript.

Reviewer 4 Report

After read the authors’ response, I think they did not give out a right enough insight for what problem existing in the current-art their manuscript wants to resolve. There is lack of the substantial analysis for what rational or new reasons to design such different topographies (anisotropic and isotropic) to observe the differentiation of nerve cells, also lack of new insightful analysis on why such different topographies could induce the different effects on the differentiation of nerve cells, and also lack of using even basic microfluidic parameters such as volume flow-rates or shear-stresses to observe the due effects. Therefore, this modified manuscript is still surface on and did not give fresh results on such similar topic. In addition, the line72-73 is not good English express, and the line117-128 is somewhat wordy for the material selection reason.

Author Response

Dear reviewer,

Thank you very much for your comments on the manuscript. The manuscript has been revised and some modifications have been made according to the comments. The responses to each point are listed as following.

                                                                                                              Sincerely,

                                                                                  Jinbo Wu and Xinghua Gao  

Point 1What problem their manuscript wants to resolve.

Response 1: Thanks for your comments. The main problem explored in this manuscript is the influence of biophysical factors of different topographical cues on nerve cell differentiation. Using these topologies, we can effectively simulate the 3D environment of the human body in vitro and show that 3D constraints could be used to guide axonal extension. Future work may increase the complexity of the microchannel, involving more complex networks of more neuronal subtypes and provide the basis for the in vitro study of Alzheimer's disease, Parkinson's disease, and other neurological disorders.

Point 2There is lack of the substantial analysis for what rational or new reasons to design such different topographies (anisotropic and isotropic) to observe the differentiation of nerve cells,and also lack of new insightful analysis on why such different topographies could induce the different effects on the differentiation of nerve cells, and also lack of using even basic microfluidic parameters such as volume flow-rates or shear-stresses to observe the due effects.

Response 2: Thanks for your comments. It was analyzed in this paper “the results showed that PC12 cells grown on different microstructures could differentiate into neurons and exhibited increasing cell body size, sprouting neurites and the formation of a neural network. In this, compared with flat group (F group), the axon length of PC12 cells increased significantly in the S, P60, P90 and C135 groups. The C45 and C90 groups may be affected by the bending angle of the polyline microstructure, the axon length did not become significantly longer, and C45 was also significantly shorter and only 20.3±2.8 µm. This revealed that the axon could rotate around the corner within the range of 45°to 180° of the measured angles on anisotropic patterns; the larger this angle, the more pronounced the axon growth was (S: 84.0±2.9 µm > C135: 54.8±3.8 µm > C90: 48.1±2.5 µm > C45: 20.3±2.8 µm).” and “PC12 cell differentiation also showed a certain cell orientation and the ability to form neural networks. The axon extension on C45, C90, C135 and S structures tended to occur in the direction of the topography itself. Conversely, the neurite extension on P60 and P90 structures was relatively more random, although most of the axons turned and extended around a pillar. Among them, the cell axons of the S group obviously showed the orientation of growing along the straight microstructure, while in the P60 and P90 group the cells were similar with the F group and could form the neural network. It might be because there was much space between the micropillars in the P60 and P90 group, and the physical constraints have little effect on the direction of cell axon growth. This was also reflected in the similar axon length of P60 (68.3±4.1 µm) and P90 (63.8±3.6 µm) and suggested that the axon length was not affected by the angle between the micropillars.”.

In this experiment, after mixing drug with the medium and after mixing the growth factor with the medium, pipetting guns were used to slowly add one side of the microchannel (the cells were seeded on the other side), and the above steps were repeated every 8 hours. So there was no discussion about the impact of flow on diffusion and mass transport.

In page 9, line 318-320,we have added“After mixing drug with the medium and after mixing the growth factor with the medium, pipetting guns were used to slowly add the right side of the microchannel, and the above steps were repeated every 8 hours.”.

Point 3The line72-73 is not good English express, and the line117-128 is somewhat wordy for the material selection reason.

Response 3: Thanks for your comments. The manuscript has been modified.

In page 2, line 74-76,We have changed “we designed and manufactured included isotropic pillars of different sizes and anisotropic micrometer gratings of different angles.”to “we designed and manufactured a series of different microstructure topographical cues for inducing neuronal differentiation of cells in vitro, which were essentially with different topography, sizes and structural complexities.”.

In page 3, line 120-128, the original content of the manuscript was changed to “Microfluidic chip materials include cyclic olefin copolymer (COC), Polymethylmethacrylate (PMMA), PDMS among others [24]. PDMS as a microfluidic chip material shows certain characteristics: high thermal stability, high biocompatibility, good optical properties, chemically inert and so on [25-26]. Considering the advantages of PDMS, we finally used PDMS as the microfluidic chip material in this experiment. Similarly, there are many types of processing methods for microfluidic systems, such as injection molding, laser ablation, molding and 3D printing, etc. [27-28]. Each method has its own advantages and disadvantages. For example, 3D printing is flexible and fast in the chip processing process. However, the choice of materials for 3D printing is very limited [29].”.